# Cardiotoxicity of Tyrosine Kinase Inhibitors in Philadelphia-Positive Leukemia Patients

**Adriatik Berisha [1,2], Angelo Placci [3] and Pier Paolo Piccaluga [4,5,*]**

1 Department of Hematology, University Clinical Center of Kosovo, 10000 Prishtina, Kosovo
2 School of Medicine, University of Zagreb, 10000 Zagreb, Croatia
3 Cardiology Unit, Cardio-Thoracic-Vascular Department, Parma University Hospital, 43126 Parma, Italy
4 Biobank of Research and Institute of Hematology and Medical Oncology "L. and A. Seràgnoli", IRCCS Azienda Opedaliera-Universitaria S. Orsola-Malpighi Hospital, 40137 Bologna, Italy
5 Department of Medical and Surgical Sciences, University of Bologna School of Medicine, 40137 Bologna, Italy
* Correspondence: pierpaolo.piccaluga@unibo.it; Tel.: +39-0512144043; Fax: +39-0512144037

**Abstract:** In the past twenty years, tyrosine kinase inhibitors (TKIs) have substantially changed the therapeutic landscape and the clinical outcome of several cancers, including Philadelphia-chromosome positive chronic myeloid leukemia and acute lymphoblastic leukemia, chronic eosinophilic syndromes, gastrointestinal stromal tumors, and others. Despite the obvious advantages offered in terms of efficacy and the overall safety profile, this new class of agents presents novel side effects, sometimes different from those induced by conventional chemotherapy. Among others, the potential cardiac toxicity, characterized by possible arrhythmias and the highest rates of cardiac ischemic disease and heart failure, were predominantly investigated. In this article, the authors review the most significant evidence in this regard, highlighting the overall benefit of TKI usage and the need for careful monitoring, especially in elderly patients.

**Keywords:** tyrosine kinase inhibitors; chronic myeloid leukemia; acute lymphoblastic leukemia; *BCR:ABL1*; cardiotoxicity; ponatinib; dasatinib; nilotinib; imatinib

## 1. Background

Cancer treatment has historically encompassed, besides surgery, chemotherapy and radiotherapy. Both these approaches aim to target and kill proliferating cells, interfering on various levels with correct DNA replication. Despite potential benefits and efficacy, as in the case of many hematological malignancies, such as acute leukemias and lymphomas, targeting proliferating cells is relatively unspecific; therefore, side effects are common, as most non-neoplastic diving cells are affected, including bone marrow cells and mucosal epithelia. Indeed, as chemotherapy resistance occurs frequently, higher doses were adopted, which overcame resistance in some instances. Furthermore, high-dose chemotherapy (associated or not with chemotherapy) were also used as conditioning regimes for stem cell transplantations. However, as expected, the higher efficacy was paralleled by higher toxicity.

Furthermore, specific chemotherapy agents can also present a peculiar toxic profile on non-dividing cells as in the case of anthracyclines, which induce dose-dependent cumulative toxicity on myocardial cells.

In order to increase the clinical efficacy and reduce the overall toxicity in the past two decades, so-called targeted agents were introduced as anticancer treatment, leading to substantial benefits in many cases. Such approaches are aimed to more or less selectively target neoplastic cells by inhibiting specific cellular pathways known to be relevant for cancer cell survival. At the same time, as their effects are selectively targeted to cancer cells, these new drugs came with the promise of significantly reduced side effects, therefore named "smart" or "intelligent" chemotherapy.

The most striking examples of targeted therapies for cancer consist of anti-leukemia therapies; specifically, all-trans-retinoic acid (ATRA) for acute promyelocytic leukemia and imatinib mesylate for chronic myeloid leukemia (CML). The latter was the first-in-class tyrosine kinase inhibitor (TKI) and profoundly changed the history of CML [1]. CML, in fact, could be cured only by allogeneic stem cell transplantation, a procedure at that time largely limited by toxicity in older patients and donor availability in younger patients. Imatinib and second/third-generation TKIs, by contrast, were able to induce long-term molecular complete remission in a remarkably high percentage of patients. Of note, this was paralleled by an excellent toxicity profile, which allowed long-term treatment in an outpatient setting.

However, TKIs can induce, as any other targeted regimen, mild to serious side effects, including cardiotoxicity. Since most CML patients are 65 years or older, this is particularly relevant, as many patients may present a concomitant cardiovascular disease together with CML.

Recently, a large Italian study reporting the real-life data of a cohort including 656 CML patients in the chronic phase and treated with second- or third-generation TKIs indicated that cardiovascular (CV) events significantly affected overall survival [1]. In particular, 12/37 deaths were related to CV complications. The 15-year CV mortality-free survival was 93%. Patients aged $\geq$65 years showed a significant lower CV mortality-free survival ($72.1 \pm 13.1\%$ vs. $95.8 \pm 2.7$, $p < 0.001$). In multivariate analysis, age $\geq$ 65 years ($p = 0.005$) and a positive history of AOEs ($p = 0.04$) were confirmed to be significantly associated with a lower CV mortality-free survival [1]. These data highlighted the need for careful clinical monitoring and adequate prevention strategies.

In this article, the authors review the main evidence about TKI-induced cardiotoxicity (summarized in Table 1), focusing on the main clinical trials that led to drug approvals, reviews, and real-life data.

**Table 1.** Tyrosine kinase inhibitors and type of reported cardiotoxicity.

| Agent | Tyrosine Kinase Targets | Type of Cardiac Toxicity |
|---|---|---|
| Imatinib mesylate | BCR-ABL1, KIT, PDGFRs | CHF, LVEF depression |
| Dasatinib | BCR-ABL1, KIT, PDGFRs, SRC Family | QT prolongation, peripheral edema, pericardial effusion |
| Nilotinib | BCR-ABL1, KIT, PDGFRs | QT prolongation |
| Ponatinib | BCR-ABL1, KIT, PDGFRs | MI, CHF, cardiomyopathy, vascular occlusion |
| Sunitinib | VEGFR 1–3, RET, PDGFRs, KIT, FLT3, CSF1R | Hypertension, LVEF depression, CHF, MI |
| Sorafenib | VEGFR 2&3, KIT, PDGFRB, FLT3, RAF1, BRAF | Acute coronary syndrome, MI, hypertension |
| Lapatinib | EGFR, ERBB2 | Asymptomatic LVEF depression |

CHF: Chronic heart failure; LVEF: Left ventricle ejection fraction; MI: Myocardial infarction.

## 2. Imatinib Mesylate-Induced Cardiotoxicity

Imatinib mesylate (formerly STI571, Glivec®, Novartis, CH) is an oral selective inhibitor of abelson (ABL1), platelet-derived growth factor receptor (PDGFR) alpha and beta, and KIT tyrosine kinases (TK) [2]. The mechanism of action of imatinib is a function of its ability to target proteins involved in the pathogenesis of specific cancer types. In particular, imatinib is the first-line agent for the treatment of Philadelphia (Ph) chromosome-positive (Ph-pos, that is, *BCR-ABL1* translocation-positive) chronic myeloid leukemia (CML) [3], because it inhibits the TK activity of the leukemogenic *BCR-ABL1*-encoded fusion protein [4]. In addition, it is currently licensed for the treatment of Ph-pos acute lymphoid leukemia (Ph-pos ALL), hypereosinophilic syndrome (HES) and/or chronic eosinophilic leukemia

(CEL) with *FIP1L1-PDGFRA* rearrangement, MDS/MPD associated with *PDGFR* gene rearrangements, and gastro-intestinal stromal tumors (GISTs) [2].

Soon after its introduction into clinical practice, the enthusiasm for the clinical success of imatinib was tempered by the observation of its possible cardiotoxicity, documented both in a clinical setting and in experimental models [5]. In particular, Kerkelä et al. reported ten individuals with CML who developed severe congestive heart failure while being treated with imatinib [5]. In addition, they showed that imatinib-treated mice developed left ventricular contractile dysfunction and concluded that cardiotoxicity is an unanticipated side effect of ABL1 inhibition by imatinib. Furthermore, in the same issue of *Nature Medicine*, an accompanying *News and Views* article pointed out that TK inhibitors are plagued by cardiotoxic side effects [6].

Ribeiro and colleagues investigated the possible cardiotoxicity of imatinib in a prospective study [7]. They included 103 CML patients treated with imatinib and 57 patients with chronic myeloproliferative disorders not treated with imatinib in order to evaluate its cardiotoxicity. There was no statistical difference regarding cardiac symptoms and signs, BNP levels, and echocardiographic measurements for the imatinib and control groups, except for peripheral edema, which was more frequent in the imatinib group but not however related to cardiac dysfunction. Four patients in the imatinib group presented a B-type natriuretic peptide (BNP) level >100 pg/mL, but only one of them with depressed left ventricular ejection fraction (LVEF). Overall, imatinib was not related to the systematic deterioration of cardiac function, although the possibility of isolated cases of cardiotoxicity could not be excluded. Indeed, this study had some limitations as the sample size probably did not allow the recognition of very rare adverse effects. However, this study was prospective and the included patients received imatinib for a long period (at least 24 months) [7].

Notably, these results are consistent with those reported by several groups [8–14]. In particular, 6 reports including 5586 patients affected by Ph$^{-pos}$ CML/ALL (N = 4123), GIST (N = 1144), or other rare malignant disease (N = 319) cured at different European and American Institutions. History and clinical parameters, with special attention to symptoms of heart failure or cardiac disease were evaluated. In one study, N-terminal pro BNP and serum cardiac troponin were considered [14]. Overall, very few patients (0–0.6%) had cardiac events, including congestive heart failure and cardiac death. Notably, in all instances, these patients were either of advanced age or presented with personal history of cardiac disease before entering imatinib treatment.

The mechanism through which imatinib might affect myocardial cells is not well understood. In fact, the exact role of ABL1 cardiomyocytes is not clear since it mediates oxidant stress-induced death in fibroblasts and is protective in osteoclasts [15]. If analogous protection is offered to cardiac cells, its inhibition might certainly favor oxidative-stress induced myocardial injury [16]. In addition, experimentally, it was shown that JNK activation may be responsible for the cardiac toxicity observed with imatinib, and the inhibition of the JNK pathway markedly reduces the collapse of the mitochondrial membrane potential and cell death [16,17]. Indeed, JNK inhibition (achieved, for example, by a methylated form of imatinib) was shown to be associated with significant cardiotoxicity reduction without affecting the therapeutic effects [16,18].

Even greater attention to their potential cardiotoxicity was given to second- and third-generation TKIs, including nilotinib, dasatinib, and bosutinib. In fact, they come with either more powerful activity on BCR-ABL1 and/or more pronounced off-target activity.

### 3. Dasatinib-Induced Cardiotoxicity

Dasatinib (BMS-354825) is a second-generation oral, multi-targeted tyrosine kinase inhibitor that targets BCR-ABL and SRC kinases. Dasatinib has 325-fold greater potency vs. imatinib in cell lines transduced with wild-type *BCR-ABL1* and is active against 18 out of 19 *BCR-ABL1* mutations tested with imatinib resistance. Dasatinib inhibits BCR-ABL1, the SRC family of kinases (SRC, LCK, YES, and FYN), KIT, EPHA2, and PDGFRs at nanomolar concentrations. Dasatinib was active in vitro in several imatinib-sensitive or imatinib-

resistant leukemic cell lines and inhibited the growth of CML and acute lymphoblastic leukemia cell lines overexpressing BCR-ABL1.

While dasatinib potently inhibits BCR-ABL1, it is not a highly selective kinase inhibitor and may have off-target effects. Therefore, the risk of side effects, different or more pronounced than observed with imatinib, has to be faced. In this regard, Dr. Brave and colleagues showed that the acute and chronic toxicities of dasatinib in pre-clinical studies included cardiovascular toxicities, including QT prolongation in vitro and increased systolic, diastolic, and arterial blood pressure in animals. In addition, vascular and cardiac fibrosis, cardiac hypertrophy, myocardial necrosis, hemorrhage of the valves, ventricle, and atrium, and cardiac inflammation were noted in toxicology studies [19]. Consistently, Wong et al. showed that dasatinib could actually prolong QT interval of patients a from review of clinical profiles of dasatinib-treated patients with imatinib-resistance [20]. Finally, the DASISION trial study, randomly comparing dasatinib vs. imatinib in the treatment of naïve CML patients, indicated that the risk of arterial ischemic events was 5% during dasatinib treatment at a daily dose of 100 mg [21].

Overall, although dasatinib can be considered a relatively safe drug, QT monitoring before and during treatment are recommended.

## 4. Nilotinib-Induced Cardiotoxicity

Nilotinib (Tasigna) is a BCR-ABL1 tyrosine kinase inhibitor that is approved for the treatment of patients with Philadelphia chromosome-positive leukemias (CML and ALL) in first or subsequent lines. Interestingly, in contrast to imatinib, the cellular uptake of nilotinib was independent of active transporter expression, so that systemic exposure is likely to be more closely related to patient response. Dr. Kim and colleagues reported on the clinical cardiac safety profile of nilotinib [22]; in this study, considering 81 patients with a median treatment duration of 26 months, they found that 16 out of the 81 patients (20%) had new electrocardiographic changes [22]. In this regard, Dr. Wendy et al. found, in an in vitro model, that nilotinib at pharmacologically relevant concentrations might be less toxic to cardiac cells than imatinib and dasatinib. In fact, despite inducing decreased leukemic cell viability, unlike imatinib and dasatinib, nilotinib did not significantly affect the mitochondrial membrane potential integrity under identical experimental conditions [23]. Therefore, the molecular mechanisms of the observed cardiotoxicities in the clinical setting may be different differ between imatinib and nilotinib. Indeed, besides direct effects on myocardial cells, a multicenter case series study suggested that a subset of patients treated with nilotinib was developing severe peripheral arterial disease (PAD). In the original study, 11 (6.2%) of 179 patients had severe PAD, with 8 patients requiring angioplasty, 8 patients requiring stent placement, and 4 patients requiring subsequent lower limb amputation [24].

Despite that, long-term follow-up studies revealed an overall safe profile of nilotinib, with the remarkable clinical efficacy largely exceeding the risk of toxicities, including cardiac toxicity [25].

## 5. Ponatinib and Cardiotoxicity

Ponatinib is a potent TKI, approved for the treatment of Ph$^{-pos}$ leukemias resistant to imatinib, due to its ability to inhibit mutated forms of ABL1, including those carrying the T315I mutation.

In a phase II trial, 18/449 patients died during the study [26,27]. Among other side effects, cardiovascular ones were well-documented. In particular, vascular occlusion occurred in 23% of cases, and some patients experienced fatal myocardial infarction [26,28–30]. Furthermore, treatment with ponatinib was also associated with cardiomyopathy and congestive heart failure (CHF) [26,28–31]. Noteworthy, such cardiac events were observed at any age, in the presence or absence of cardiovascular risk factors. Overall, the risk of severe cardiovascular events was estimated to be around 9% in patients receiving ponatinib [26].

Singh and colleagues deeply explored the possible molecular mechanisms underlying ponatinib toxicity on myocardial cells [26]. They found that selective FGFR inhibition con-

tributes minimally to the ponatinib-induced cardiotoxicity. Conversely, they showed that ponatinib inhibits cardiac pro-survival signaling pathways AKT and extra-cellular-signal-regulated kinase (ERK) and induces cardiomyocyte apoptosis [26]. Of note, strengthening AKT and ERK signals through the administration of neuregulin-1β was effective in preventing ponatinib-induced cardiomyocyte apoptosis [26]. Importantly, in the case of aorta and megakaryocytes, the pro-apoptotic effect of ponatinib leads to a prothrombotic state and hyperactive platelets [32].

Overall, it should be considered that ponatininb has a very broad spectrum of TKI inhibition [33], and this wide spectrum of activity leads to its toxicity.

## 6. Other TKIs and Cardiotoxicity

Beside the TKIs used for the treatment of Ph^-pos leukemias, other TKIs have been successfully introduced in the treatment pipelines of several solid cancers, including among others sorafenib, sunitinib, lapatinib, gefitinib, and erlotinib. Sorafenib is a multi-target TKI, inhibiting KIT, PDGFRB, vascular endothelial growth factor receptors (VEGFR) 2 and 3, BRAF, RAF1, and FLT3. It is used for several malignancies, including renal cell cancer and hepatocellular carcinoma. Recently, it was also used in peripheral T-cell lymphomas [34]. Despite not common, sorafenib is well-known to induce acute coronary symptoms, including myocardial infarction, in about 3% of patients [16,35]. Both RAF1 and VEGFR inhibition have been postulated to be responsible for such events: on the one hand, RAF1 protects myocardial cells from oxidative stress; on the other hand, the activity on VEGFRs may lead to the reduction of capillary permeability and increased pressure load, leading to hypertrophy of the heart and subsequently congestive heart failure [16].

Sunitinib targets VEGFRs 1–3, KIT, RET, PDGFRA/B, FMS-related tyrosine kinase 3 (FLT3), and the colony stimulating factor 1 receptor (CSF1R). It is approved for first- and second-line treatments of solid cancers, including renal cell carcinoma and GIST. Notably, a remarkable percentage of patients receiving this agent sunitinib developed hypertension, left ventricular dysfunction, and other cardiac events [16]. Therefore, careful attention and cardiologic monitoring is recommended in all patients treated with sunitinib.

Lapatinib induced a decrease in LVEF > 20% in some instances when used for treating metastatic breast cancer, leading to treatment suspension. However, for unknown reasons, the effect was definitely less common than with trastuzumab, the monoclonal antibody targeting the same receptor, ERBB2 [16].

By contrast, no significant cardiotoxic effects have been reported as far as the epidermal growth factor receptors inhibitors (gefitinib and erlotinib) are concerned [16].

## 7. Arterial Disease and TKIs

In a recent meta-analysis, Dr. Mulas and colleagues investigated the role of second- and third-generation TKIs in inducing hypertension in CML patients [36]. In their pooled analysis, the authors found that hypertension is a common complication of TKIs; in particular, the incidence was 10% for all new-generation TKIs, with a remarkably higher prevalence in patients receiving ponatinib (17%). The comparison with the first-generation imatinib confirmed that ponatinib was associated with the highest risk (RR 9.21; 95% CI; $p = 0.0002$). In addition, it showed that also nilotinib was associated with a significantly increased risk of hypertension (RR 2; 95% CI; $p = 0.0002$) [36].

In a large cohort of 192 patients with hypertension at CML diagnosis who were treated with second- or third-generation TKIs, the efficacy of renin-angiotensin system (RAS) inhibitors [angiotensin-converting enzyme inhibitors (ACEi) and angiotensin-II receptor blockers (ARBs)] in the prevention of AOEs was evaluated and compared with other drug classes [37]. The 5-year cumulative incidence of AOEs was about 33%. Patients with SCORE $\geq$ 5% (high/very-high) showed a significantly higher incidence of AOEs ($33.7 \pm 7.6\%$ vs. $13.6 \pm 4.8\%$, $p = 0.006$). The AOE incidence was significantly lower in patients treated with RAS inhibitors ($14.8 \pm 4.2\%$ vs. $44 \pm 1\%$, $p < 0.001$, HR = 0.283). The difference in the low and intermediate Sokal risk group was confirmed, but not in the

high-risk group, where a lower RAS expression has been reported. The authors concluded that RAS inhibitors may represent an optimal treatment in patients with hypertension and CML, treated with second- or third-generation TKIs [37].

Interestingly, CML patients with a previous history of arterial occlusive events (AOE) and treated with second- or third-generation TKIs have a significant risk of recurrent events even in the presence of secondary prophylaxis [38]. Based on such evidence, individualized treatment is needed to optimize secondary prevention. In a study by Caocci et al., it was shown that the 60-month cumulative incidence rate of recurrent AOEs was about 48%. Despite a history of AOE, 10 patients (16%) included in the study were not receiving secondary preventive treatments such as an antiaggregant. Overall, seventeen recurrent AOEs were observed, while three cardiovascular-related deaths were reported [38]. Patients receiving nilotinib and ponatinib showed a higher incidence of recurrent AOEs ($76.7 \pm 14.3\%$ and $64 \pm 20.1\%$, respectively) than those treated with dasatinib and bosutinib ($44 \pm 24.2\%$ and $30.5 \pm 15.5\%$, respectively) ($p = 0.01$). Only treatment with a second- or third-generation TKI given as second or subsequent line therapy showed a significant association with an increased incidence of recurrent AOE ($p = 0.039$) [38].

## 8. Conclusions

The introduction of imatinib and other TK inhibitors has prompted important changes in the management of CML and other diseases. Particularly, as compared to conventional chemotherapy, not only was treatment efficacy significantly enhanced, but the toxicity profile was remarkably reduced. However, although targeted therapies are overall considered less toxic and better tolerated by patients compared with older chemotherapy drugs, certain complications can be very serious [16]. In addition, TKI are usually administered for prolonged periods of time, thus facilitating the occurrence, even later, of adverse events. Cardiac toxicity, in particular, may range from asymptomatic subclinical abnormalities such as electrocardiographic changes and left ventricular ejection fraction decline to life-threatening events such as congestive heart failure and acute coronary syndromes [16,39]. Overall, the available data on cardiac toxicity do not weaken the overall risk–benefit profile of TKIs, although careful evaluation and monitoring is warranted in elderly patients as well as when pre-existing cardiac dysfunction is documented [1,40–43].

**Funding:** This work was supported by BolognAIL (2020, Piccaluga), RFO DIMES (2018–2022 Piccaluga), and FIRB Futura 2011 RBFR12D1CB (Piccaluga).

**Institutional Review Board Statement:** Not applicable.

**Informed Consent Statement:** Not applicable.

**Data Availability Statement:** Not applicable.

**Acknowledgments:** This manuscript is in memoriam of Prof. Michele Baccarani (1942–2021).

**Conflicts of Interest:** The authors declare no conflict of interest.

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
