# Peer review of "Cardiotoxicity of Tyrosine Kinase Inhibitors in Philadelphia-Positive Leukemia Patients"

_hemato, doi:10.3390/hemato4010007_

Round 1
Reviewer 1 Report
Overview
The authors should "tone down" their assessment of cardiovascular toxicity of TKIs. I suggest they remove the "much ado about nothing" from the title. The manuscript does not end with that conclusion. Although they are not impressed with the evidence of these agents in specific cardiac effects, some of these agents, e.g., ponatinib, are quite toxic to the cardiovascular system. One does not want to leave the reader with the impression that these agents are benign to the cardiovascular system, in general.
Specific
p. 1, line 39. "targeted" agents, not target agents.
p. 4., line 153. "the" should be changed to "they".
p. 4 bottom paragraph on ponatinib. The authors need to emphasize that ponatinib has a very broad spectrum of TKI inhibition. See Peng and Schmaier, Int. J. Mol Sci. 21:6556, 2000. This wide spectrum of activity leads to its toxicity.
p. 4, lines 183-184. A major mechanism of ponatinib toxicity is the induction of cellular apoptosis. In the case of aorta and megakaryocytes, this effect leads to a prothrombotic state and hyperactive platelets. See Merkulova A et al. Blood Advances 3:2312, 2019. Although these effects are not on the heart, pre se, they are not "much ado about nothing". One does not want to leave the reader the impression that these agents do not disrupt the cardiovascular system, in general.
Author Response
Dear Sir,
thank you very much for your comments, that we found useful indeed. We modified the manuscript as suggested. Please find the point-by-point reply attached.
Best regards
Pier Paolo Piccaluga

Reviewer 2 Report
Adriatik Berisha, Angelo Placci and Pier Paolo Piccaluga in Cardiotoxicity of Tyrosine Kinase Inhibitors in Philadelphia Positive Leukemias Patients: Much ado about Nothing? (review) pay attention on the huge TKis efficacy on CML, emphasizing how these molecules changed the CML history in this era, respect to chemotherapy and stem cell transplantation. This is was twentytwo years ago. Now we have five TKIs, asciminib is coming, hematologists change the CML history if know these molecules. The kwnoledge of TKIs toxicity prophyle is the only important field to treat CML.
The references showed are too old. Nilotinib and Dasatinib were born in AIFA in 2009, Ponatinib in 2013. So, in the last ten years hematologists who treat CML know the cardiotoxicity of these drugs because they consider it on their patients.
Title is very attractive but could be misinterpreted.
Much ado about nothing is too strong. The question mark is not enough...
Cardiotoxicity in......: the "TKIs expertise" is the hematologist requirement to treat CML in 2023.
Giovanni Caocci, Italy, wrote 5 manuscripts about cardiotoxicity and cardiovasculat features in CML patients treated with TKIs. These data are from Italian Centers. Attached the list of manuscripts in PubMed.

Author Response

(The authors gave the same response as above.)

Round 2
Reviewer 2 Report
Thank you for your reply
Then, Reviewers agree to adequacy of the title. Much ado for nothing is not true. Not really much ado for nothing is not clear for the readers.
Editor can help you. I would suggest:
cardiotoxicity....: don't forget background noise
Author Response
Thank you for your reply
We simplified the title deleting the much ado about nothing
Best regards